# Hypothermia in Trauma

**DOI:** 10.3390/ijerph18168719

**Published:** 2021-08-18

**Authors:** Michiel J. van Veelen, Monika Brodmann Maeder

**Affiliations:** 1Eurac Research, Institute of Mountain Emergency Medicine, 39100 Bolzano, Italy; monika.brodmannmaeder@insel.ch; 2Department of Emergency Medicine, University Hospital Bern and Bern University, 3010 Bern, Switzerland

**Keywords:** hypothermia, trauma, hemorrhage, lethal triad, coagulation, traumatic brain injury, cardiac arrest, extracorporeal life support, ECLS, emergency preservation resuscitation

## Abstract

Hypothermia in trauma patients is a common condition. It is aggravated by traumatic hemorrhage, which leads to hypovolemic shock. This hypovolemic shock results in a lethal triad of hypothermia, coagulopathy, and acidosis, leading to ongoing bleeding. Additionally, hypothermia in trauma patients can deepen through environmental exposure on the scene or during transport and medical procedures such as infusions and airway management. This vicious circle has a detrimental effect on the outcome of major trauma patients. This narrative review describes the main factors to consider in the co-existing condition of trauma and hypothermia from a prehospital and emergency medical perspective. Early prehospital recognition and staging of hypothermia are crucial to triage to proper care to improve survival. Treatment of hypothermia should start in an early stage, especially the prevention of further cooling in the prehospital setting and during the primary assessment. On the one hand, active rewarming is the treatment of choice of hypothermia-induced coagulation disorder in trauma patients; on the other hand, accidental or clinically induced hypothermia might improve outcomes by protecting against the effects of hypoperfusion and hypoxic injury in selected cases such as patients suffering from traumatic brain injury (TBI) or traumatic cardiac arrest.

## 1. Introduction

Major trauma, defined as an Injury Severity Score (ISS) of 15 or greater, accounts for 10% of all deaths worldwide [1]. Hypothermia, defined as a body temperature of 35 °C or less, is present in up to two-thirds of these patients upon admission to the emergency department (ED) [2]. Hypothermia is currently estimated to occur less frequently with modern hemostatic, goal-directed resuscitation with warm fluids [3]. Risk factors for hypothermia in trauma include the severity of injury, (prehospital) anesthesia or intubation, low environmental temperature or wet clothing, and administration of cold fluids [4,5]. Therefore, hypothermia in trauma patients can result from environmental exposure, the injuries sustained leading to hypovolemic shock due to hemorrhage, or medical interventions, and it is often the result of a combination of these factors. Hypothermia in trauma patients leads to higher in-hospital mortality, higher transfusion requirements, and longer length of stay [6,7,8]. Mortality increases with the degree of hypothermia in severely injured patients [6,8]. Since the late 1970s, studies have demonstrated a relationship between hypothermia and coagulation [9,10], but only in 1999 did the expression of the “lethal or trauma triad of death” appear in scientific articles [11]. It describes the combination of acidosis, hypothermia, and coagulopathy, which is a condition that causes a substantial rise in mortality in severe trauma [12]. This report will overview the physiological process in hypothermic trauma patients focusing on the lethal triad of coagulopathy, hypothermia, and acidosis. Furthermore, it will describe hypothermia staging, its implications, and treatment options. It also describes the effect of hypothermia on clinical outcomes in trauma patients and patients with traumatic brain (TBI) injury. Additionally, it further explains the role of clinically induced hypothermia in traumatic cardiac arrest. 

## 2. Coagulation

The combination of hypothermia, acidosis, and coagulopathy has become well-known as the trauma triad of death, diminishing the chances of survival in severely injured patients [11,12]. The three factors are highly interrelated and create a vicious circle if not addressed in a very early stage of trauma management. Acidosis is usually caused by diminished organ perfusion in shock, primarily hemorrhagic shock in trauma. It directly influences thrombin generation, which results in impaired coagulation function [13]. Hypothermia is the next factor that affects blood clotting in different ways. All chemical reactions are temperature-dependent. This is also the case with blood clotting. Therefore, it seems evident that the function of the humoral coagulation cascade deteriorates with decreasing temperatures. However, several studies have also shown that platelet function is influenced by temperature [14,15]. Thrombin seems to play a major role in platelet dysfunction through a platelet adhesion defect in the presence of hypothermia [16,17]. Fibrinolysis appears to play a minor role in the development of coagulopathy [15]. Unfortunately, all lab tests assessing coagulation are performed in normothermia. A correction in the presence of hypothermia is made purely by calculation and therefore does not necessarily correspond to the actual coagulation function [17,18]. As early as the 1990s, it was proposed to use viscoelastic hemostatic assays (VHA) such as thromboelastography (TEG) and rotational thromboelastometry (ROTEM) to evaluate the entire coagulation system [19]. Unfortunately, it did not lead to a breakthrough in the diagnosis of coagulation disorders in the presence of hypothermia [20,21].

## 3. Clinical Management

### 3.1. Staging

Prehospital staging of hypothermia is based on clinical status, the presence of vital signs and, if available, body temperature measurement. The Swiss staging model describes stages I–IV with ranges of stage 1 from <35–32 °C, for stage 2, <32–28 °C; for stage 3, <28–24 °C; and for stage 4, below 24 °C [22]. The Swiss staging model is based on observations of the vital signs at presentation and allows core temperature to be estimated from clinical indicators only. Patient factors such as head injury, intoxication, and profound shock in trauma can influence clinical findings [23,24], and therefore, definitive assessment of the severity of hypothermia in trauma patients requires an accurate temperature measurement. A recent evaluation of the Swiss staging model found that the temperature of patients was overestimated in 18% of cases, potentially leading to under-treatment due to an underestimation of the risk of cardiac arrest [25]. Therefore, measuring temperature is a helpful addition but can be challenging in a prehospital setting due to environmental conditions. The insertion of an esophageal probe is the most accurate measurement of a core temperature in the prehospital setting. It is reserved for intubated patients. Measuring core temperature with special urinary catheters in the bladder or central IV lines is reserved for clinical management, and rectal probes provide a delayed measurement and are impractical to insert prehospitally [23]. Epitympanic probes are in use but not widely adopted due to limited availability [26]. Epitympanic temperature measurements correlate well with the core temperature [27]. It has been used successfully in cold, prehospital environments. However, to get a reliable measurement, it is essential that the probe in the ear canal is insulated and the ear canal is free of debris such as water and snow [24,28,29]. Correct identification of stage 3 and 4 hypothermia in major trauma patients might be lifesaving as they likely require invasive active rewarming in the form of extracorporeal life support (ECLS) due to cardiac instability, which is only available in selected hospitals [28,30].

### 3.2. Treatment

Prevention of further cooling is essential and should be started in the prehospital phase for all trauma patients. Passive methods preventing further heat loss entail removing wet clothing or directly applying a vapor barrier, applying insulation foils and blankets, and increasing the ambient temperature to allow the patient to self-rewarm [31]. Active external rewarming can be achieved using heat packs and forced warm air [22,23]. Heat transfer is most efficient when heat packs are applied to the axillae, chest, and back [32]. Active internal rewarming can be initiated in the emergency department through warmed intravenous infusions, peritoneum, bladder, or thoracic lavage with warm fluids and extracorporeal blood rewarming through ECLS [23]. Extracorporeal blood rewarming has been demonstrated to lead to good clinical outcomes in severe trauma patients with deep hypothermia [33,34,35]. Vascular, bladder, or thoracic access for active internal rewarming in trauma patients can be challenging due to the obstruction of immobilization devices or (suspected) injuries [36].

Care should be taken to avoid further cooling caused by medical interventions during all phases of patient care, such as assessment on the scene, during transport, evaluation in the ED, and during damage control surgery. The value of exposure during physical exam should be weighed against the risk of hypothermia and should be performed sequentially by body region, preserving insulating clothing in the non-examined area. [26]. Resuscitation with intravenous fluids can worsen trauma-induced coagulopathy by diluting clotting factors and dislodging clots by raising blood pressure and inducing hypothermia [4]. Limiting fluid resuscitation and practicing early permissive hypotension in hypovolemic shock has a role in managing trauma patients, with the notable exception of patients with cardiac tamponade [37] and with the optimum level of targeted blood pressure still to be determined in patients with suspected raised intracranial pressure following a TBI [38].

Normothermia provides the optimum conditions for hemostasis [39]. A study from the 1960s compared the effects on mortality of administering 3 L of cold blood to 25 patients. This resulted in a cardiac arrest in 12 patients. Administering 3 L of warmed blood to a matched group resulted in a cardiac arrest incidence of only 3% [40]. Severe hemorrhage and hemodynamic instability in trauma patients are indications to start a massive transfusion protocol. In many institutions, this consists of restrictive use of crystalloids and giving plasma, platelet, and RBC products at a ratio of 1:1:1, and if available, adjusting the ratio by an early point of care VHA result [3,41]. The exact value of VHAs in (deep) hypothermic patients is yet to be determined [3,20]. In order to prevent infusion-related hypothermia, fluids and blood products should be administered via (high flow) in-line infusion warmers in the hospital setting or similarly functioning battery-operated devices in the prehospital setting [4,42]. Prehospital administration of analgesia or anesthesia may accelerate cooling due to sympathetic inhibition and vasodilation. An alternative medication choice, such as ketamine, may provide the least risk for further cooling [28].

## 4. Impact of Hypothermia

### 4.1. Major Trauma

Several reviews have assessed the effect of hypothermia on the outcome of major trauma patients and concluded it to be an independent predictor for mortality [43,44,45]. A recent systematic review included seven studies for meta-analysis and concluded that accidental hypothermia at admission was associated with significantly higher mortality in trauma patients with an OR of 5.18 (95% confidence interval (CI), 2.61–10.28) [44]. Hypothermia is a marker for a poor prognosis after hemorrhage, probably representing metabolic dysfunction [46]. However, it has been hypothesized that hypothermia could be part of the detrimental physiological effects of severe injury and hemorrhage (and thus acidosis and coagulopathy) instead of a true independent predictor for mortality [47]. Several studies have reported a higher blood transfusion requirement [48,49] or an elevated INR [43,50] in hypothermic trauma patients related to hypothermia-induced coagulopathy. A study that assessed ICU length of stay and ventilator days found a significantly longer duration of critical care dependability of surviving hypothermic trauma patients versus non-hypothermic trauma patients [6].

### 4.2. Traumatic Brain Injury

In the literature, the hypothesis of a neuroprotective effect of hypothermia on patients with TBI by stopping the biochemical and inflammatory cascade after injury has been formulated [51]. Mild hypothermia could protect against secondary brain injury through reduced excitotoxicity, oxidative stress, apoptosis, autophagy, and inflammation [52]. Animal models have produced favorable mortality and improved behavioral outcomes after being subjected to mild hypothermia [53,54]. The effects of clinically induced as well as accidental hypothermia on mortality in this patient group have been investigated. Initially, a systematic review from 2013 showed that clinically induced hypothermia was associated with a 19% reduction in the risk of death (95% confidence interval (CI), 0.69–0.96) and a 22% reduction in the risk of poor neurologic outcome (95% confidence interval (CI), 0.63–0.98) [55]. However, recent reviews agree that there is no evidence of reduced mortality or improved neurological outcome in clinically induced hypothermia [56,57]. A recent systematic review considered the effect of accidental hypothermia at admission on mortality in two studies containing patients with TBI and concomitant injuries [48,58] as well as one study analyzing patients with isolated TBI [59]. All studies reported a higher mortality rate in hypothermic than in normothermic patients with an overall OR of 2.38 (95% confidence interval (CI), 1.53–3.69) [44]. Therefore, it seems likely that the negative impact of trauma-induced coagulopathy on mortality outweighs the potential neuroprotective effects of accidental hypothermia. However, as there is currently only one study assessing the effect of accidental hypothermia in patients with isolated TBI [59], further research seems warranted for this select group.

### 4.3. Traumatic Cardiac Arrest

The outcome of traumatic cardiac arrest is meager. Until now, we have found no evidence of the benefit of hypothermia in severely hypothermic trauma patients, whose cardiac arrest is mainly caused by severe hypothermia instead of by the trauma itself. Only one case report with a favorable outcome can be found in the literature [34].

The concept of emergency preservation resuscitation (EPR) is based on gaining rapid central arterial access to a patient in traumatic arrest and inducing profound hypothermia (<10 °C tympanic temperature). Induced profound hypothermia can protect cells during ischemia and reperfusion and decrease organ damage during cardiac arrest [51]. Extending the time window for surgical management of injuries through EPR may allow patients to survive previously lethal trauma. A feasibility trial is currently assessing therapeutic hypothermia in humans suffering from traumatic cardiac arrest [60]. The study includes patients in traumatic cardiac arrest not responding to resuscitative thoracotomy with clamping of the descending aorta.

## 5. Conclusions

Early, aggressive, goal-directed temperature management of trauma patients is of utmost importance for their survival, especially in hypothermic patients with hemorrhagic shock. Early prehospital staging of hypothermia is essential in allocating the correct level of care. The prevention of further cooling and attempted reversal of accidental hypothermia in trauma patients is crucial in prehospital and hospital management phases. Coagulation disorders must be diagnosed and treated: Rewarming is crucial to address the lethal triad in trauma and might even include ECLS in selected cases.

Hypothermia in trauma patients is strongly correlated with increased in-hospital mortality, increased length of stay, and a higher need for transfusion. At this point, there is no evidence that there is a neuroprotective effect of either accidental or induced hypothermia leading to lower mortality in patients with (concomitant) TBI. It is important to note that many cited studies on outcomes are based on retrospective data, with a lack of randomized control trials. EPR is currently being trialed and might provide the opportunity to achieve hemorrhage control after resuscitation while suppressing adverse effects of cardiac arrest and could significantly affect the prognosis.

## Data Availability

Not applicable.

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
