# Peer review of "Hypothermia in Trauma"

_ijerph, 2021, doi:10.3390/ijerph18168719_

Round 1
Reviewer 1 Report
Dear Editor,
I had the opportunity to review the narrative review: “Hpothermia in Trauma”.
In general, the review is very good. It is very concise and very well written. In my opinion there are only a few things that I would like to add or change.
Page3, 1st para: Passive rewarming is not a good term. Warming is something active. Prevention of cooling is passive. Therefore, I would like to change the start of the sentence to: “Passive methods like insulation ….”
Page 3, line 122: A previous study …I think this is a very old study therefore I think it would be appropriate to call it an old study or a study from the 1960ies. Unfortunately, the study is not cited correctly. In the study patients undergoing radical cancer surgery were studied, not trauma patients. To the best of my knowledge there are no studies like that in trauma patients.
Page 3, line132: I think it would be better to call the devices high flow in-line infusion warmers. It is possible to give names and products of manufacturers but I don’t think that this is ideal.
In many parts of the manuscript all of the information given is depending on retrospective studies and articles that have only very, very limited data to support the conclusions. There are only very few RCTs and a lot of the given information is given on pathophysiological thoughts. I think the authors should stress that and add a para about the quality of these data.
Author Response
Dear reviewer,
Thank you very much for your thorough review and solid comments to improve the manuscript. We have adapted your suggestions as described below.
Kind regards,
Page3, 1st para: Passive rewarming is not a good term. Warming is something active. Prevention of cooling is passive. Therefore, I would like to change the start of the sentence to: “Passive methods like insulation ….”
Thank you for the correct observation, we have changed the wording to: Passive methods preventing further heat loss entail...
Page 3, line 122: A previous study …I think this is a very old study therefore I think it would be appropriate to call it an old study or a study from the 1960ies. Unfortunately, the study is not cited correctly. In the study patients undergoing radical cancer surgery were studied, not trauma patients. To the best of my knowledge there are no studies like that in trauma patients.
We have modified the sentence to reflect it is indeed a very dated study from the 1960s and removed the erroneous insertion of trauma.
Page 3, line132: I think it would be better to call the devices high flow in-line infusion warmers. It is possible to give names and products of manufacturers but I don’t think that this is ideal.
Adjusted
In many parts of the manuscript all of the information given is depending on retrospective studies and articles that have only very, very limited data to support the conclusions. There are only very few RCTs and a lot of the given information is given on pathophysiological thoughts. I think the authors should stress that and add a para about the quality of these data.
We agree that this is a challenge assessing this topic. We have therefore added a description of the relatively poor data on which conclusions are based in previous studies in our conclusion section.
Reviewer 2 Report
I would like to thank the editorial board for the opportunity given to review this manuscript.
I mainly have two comments to make:
1- It would be of interest if the authors can stratify the type of treatment needed based on the grade or stage of the hypothermic patient.
2- Two articles, one comprehensive review and one original multi-institutional study, have been published on this topic. I believe it will enrich even more this review article if they are included on it. Those articles are:
- Petrone P, Asensio JA, Marini CP. Management of accidental hypothermia and cold injury. Curr Probl Surg. 2014;51(10):414-31. doi: 10.1067/j.cpsurg.2014.07.004.
- Petrone P, Marini CP, Miller I, et al. Factors associated with severity of accidental hypothermia: A multi-institutional study. Ann Med Surg (Lond). 2020;55:81-3. doi: 10.1016/j.amsu.2020.04.018.
I will be glad to review this manuscript again once those changes are addressed.
Thank you again for this opportunity.
Author Response
Dear reviewer,
Thank you very much for your thorough review and solid comments to improve the manuscript. We have adapted your suggestions as described below.
Kind regards,
1- It would be of interest if the authors can stratify the type of treatment needed based on the grade or stage of the hypothermic patient.
Thank you for your remark, however in the specific context of the (severe) trauma patient (contrary to the accidental hypothermia patient) we believe there is no indication to limit an available treatment like active external rewarming, even in mild hypothermia, especially in the prehospital setting. Of course internal rewarming is reserved for stage 3-4 patients as stated in the manuscript. As we state in the conclusion: Early, aggressive, goal-directed temperature management of trauma patients is of utmost importance for their survival, especially in hypothermic patients with hemorrhagic shock.
2- Two articles, one comprehensive review and one original multi-institutional study, have been published on this topic. I believe it will enrich even more this review article if they are included on it. Those articles are:
We thank you for the advice and for pointing us towards these very relevant articles. We have decided to add a reference to the first article mentioned in the treatment overview part.